# Bank liquidity and the risk-taking channel of monetary policy: An empirical study of the banking system in China

Cong Wang[1], Lihuan Zhuang[1,2]*

**1** School of Economics, Jinan University, Guangzhou, P.R. China, **2** Guangzhou Branch of the People's Bank of China, Guangzhou, P.R. China

* lihuan_zhuang@163.com

**Data Availability Statement:** All relevant data are within the paper and its Supporting Information files.

**Funding:** YES - This work was supported by Chinese National Funding of Social Sciences

## Abstract

This paper addresses the impact of bank liquidity on risk-taking behaviour of Chinese banks, and provides evidence for a risk-taking channel of monetary policy operating through bank liquidity. By using bank-level panel data from 123 Chinese commercial banks during 2003–2018, it is found that banks facing lower liquidity risk will be encouraged to take more risk. Moreover, loose monetary policy leads to more aggressive risk-taking by reducing the bank liquidity risk, namely a liquidity risk-taking channel of monetary policy. These findings suggest that authorities should give full consideration to the influence of the monetary policy on bank risk-taking through bank liquidity channels.

## Introduction

Liquidity risk has been recognized as a significant threat to financial institutions management and financial system stability. During the recent Global Financial Crisis, financial institutions in advanced economies faced severe liquidity risk which led to the bankruptcy or restructuring of many banks. After the Global Financial Crisis, bank liquidity has become an important focus of financial regulatory reform and financial regulators in many countries have tightened bank liquidity requirements in an effort to allay concerns about liquidity risk. However, although the increase of bank liquidity is conducive to reducing liquidity risk, it is uncertain whether the liquidity requirements will reduce the risk appetite of banks and make the whole financial system more stable. It is therefore important to better understand the underlying relationship between bank liquidity and bank risk-taking in light of the global wave of banking regulatory reforms that have made banks more liquid than ever.

Although recent studies have examined the role of liquidity risk and bank risk-taking in the context of both advanced market and developing economies [1–4], yet the results have been mixed and very little attention has been paid to an emerging market economy such as China where bank liquidity remains at a high level. As shown in Fig 1, the aggregate deposit-to-asset ratio for all Chinese bank stood at an average of 75% from 2003 to 2010 and decreased to an average of 65% in recent years, yet it is considered to be high compared to banks in 18 countries with a substantial presence of Islamic banking [3]. As China's influence on the global

(18BJL116). The funders had no role in study design, data collection and analysis, decision to publish, or preparation of the manuscript.

**Competing interests:** The authors have declared that no competing interests exist.

**Fig 1. The aggregate deposit-to-asset ratio for all Chinese bank from 2003 to 2018.**

economy and finance is on the rise and it is regarded as the role model for the other developing countries in adopting the swift economic transformation and financial reformation, the role of liquidity risk and bank risk-taking in China is of great significance to other developing countries.

Moreover, the existing literature typically identifies two main notions related to bank liquidity, namely funding liquidity and market liquidity, which are mutually reinforcing and influence the degree of bank liquidity [5–7]. As monetary policy is an important factor affecting funding liquidity and market liquidity, it is necessary to consider the influence of monetary policy when examining the relationship between bank liquidity and bank risk-taking. After the Global Financial Crisis, the linkage between monetary policy and bank risk-taking has attracted increasing attention from both academia and policymakers. An number of studies have found that a prolonged period of relatively low interest rates can induce banks' excessive risk taking and financial imbalances [8–10]. Distinct from the traditional monetary transmission mechanism, Borio and Zhu [6] coin the term "risk-taking channel of monetary policy", and a number of studies have proved the existence of the risk-taking channel and explored the mechanisms it operates, such as the impact of interest rates on valuations, incomes and cash flows, incentives to "search for yield", the communication policies and reaction function of the central bank [6, 8, 11, 12]. However, previous studies on the link between monetary policy and bank risk-taking have not fully considered the impact of bank liquidity. They just consider bank liquidity as a control variable, and rarely discuss a dimension of the risk-taking channel operating through bank liquidity.

To fill this gap, we investigate the linkage between monetary policy, bank liquidity risk and risk-taking behaviour of banks in China. In this paper, we examine the effect of banks' liquidity risk on risk-taking behaviour, and the existence of a risk-taking channel of monetary policy operating through bank liquidity, namely a liquidity risk-taking channel of monetary policy.

The paper has two contributions to the existing literature. First, we provide empirical evidence to reveal the potential relationship between bank liquidity and bank risk-taking in an emerging market economy which received only limited attention in related research thus far. Second, we expand the risk-taking channel theory by introducing an additional dimension of the transmission mechanism of monetary policy through bank liquidity and prove its existence in China.

The remainder of the paper is organized as follows. Section 2 reviews the literature on the bank risk-taking and develops the hypotheses of the study. Section 3 presents data and variables used and descriptive statistics. Section 4 presents the model and methodology. Section 5 discusses the estimated results and robustness tests. Section 6 presents study summary and conclusions of the study.

## Hypothesis development

**Bank liquidity and bank risk-taking.**   Liquidity risk, as one of the important risks, contributes to banks' probability of default. Hong et al. [13] point out that, systemic liquidity risk is a major contributor to bank failures in American during the Global Financial Crisis, and an effective framework of liquidity risk management needs to target liquidity risk at both the individual level and the system level. The harm of liquidity risk has become the focus of academic and regulatory authorities. However, will strengthening bank liquidity supervision and improving bank liquidity reduce banks' risk appetite and their risk? Vazquez and Federico [14] find that higher funding stability as measured by the net stable funding ratio in the new Basel III guidelines, reduces the possibility of bank failures in American and European. However, King [15] estimates the net stable funding ratio for banks in 15 countries and finds that strategies to increase the net stable funding ratio are estimated to reduce bank profitability and increase bank risk. Consistent with this view, the theoretical predictions of Wagner [16] suggests that high levels of asset liquidity can potentially increase banking instability and the externalities associated with banking failures because higher asset liquidity makes crises less costly for banks and encourages banks to take on an amount of new risk. Similarly, the theoretical research of Acharya and Naqvi [17] also shows that low liquidity risk as a result of large amounts of deposit inflows can induces risk-taking behaviour on the part of bank managers. Following Acharya and Naqvi [17], Khan et al. [1] use the ratio of total deposits to total assets as the proxy for funding liquidity risk and find that a reduction in funding liquidity risk increases bank risk by using data for U.S. bank holding companies. Dahir et al. [2] and Smaoui et al. [3] also find that lower funding liquidity risk leads to higher bank risk-taking in BRICS countries and 18 countries with a substantial presence of Islamic banking, respectively. However, Rokhim and Min [4] find that banks with lower liquidity risk indicated by higher deposit ratios tend to take lower risks. As the effects of liquidity risk on bank risk-taking are mixed in prior literature and we know very little about the effect in China, it is necessary to further study the situation in China. Building on the prediction of Acharya and Naqvi [17] and Khan et al. [1], we test the following hypothesis.

Hypothesis 1. The risk-taking of Chinese banks has a positive relationship with bank liquidity.

## The risk-taking channel of monetary policy

Borio and Zhu [6] coin the term "risk-taking channel of monetary policy transmission" to denote the impact of monetary policy on banks' appetite for risk-taking and explain three transmission mechanisms of this channel. The first effect operates through the impact of interest rates on valuations, incomes, and cash flows. Lower interest rates boost the value of assets

and collaterals as well as incomes and profits which in turn reduce risk perceptions, increase risk tolerance and incentivize banks' risk-taking behaviour. The second effect operates through inducing the "search for yield" effect. Loose monetary policy leads to the decline of risk-free assets yield, which drives banks to increase their investment in risky assets to meet the fixed or targeted level of returns. The third effect operates through the communication policies and reaction function of the central bank. Higher monetary policy transparency reduces market uncertainty and risk premiums and encourages bank risk-taking. Based on the research of Borio and Zhu [6], some scholars have supplemented the other mechanisms of the risk-taking channel. For example, Chen et al. [18] suggest that monetary policy would affect banks' risk-taking via causing banks to adjust their leverage, its impact on the adverse selection problem and generating competing effects.

The majority of current empirical research has documented the existence of risk-taking channel. For example, Delis and Kouretas [9] conduct an empirical test of euro bank data between 2001 and 2008, which prove that low interest rate greatly increases banks' risk-taking, and the impact of interest rate on risk would be reduced for banks with higher equity capital, while the impact of interest rate on risk would be increased for banks with large proportion of off-balance sheet business. Jimenez et al. [19] use a unique micro-level data set for Spain, and find that after the monetary expansion, Spanish banks increased their loans to less creditworthy borrowers, thus increasing bank risk. Chen et al. [18] find that banks' riskiness increases when monetary policy is eased by using bank-level panel data from more than 1000 banks in 29 emerging economies during 2000–2012.

However, previous studies on the link between monetary policy and bank risk-taking have not fully considered the impact of bank liquidity. Although Borio and Zhu [6] suggest that the link between liquidity and risk-taking can add to the strength of the monetary policy transmission mechanism-a sort of "liquidity multiplier", they don't regard bank liquidity as a transmission mechanism of risk-taking channel. Nguyen and Boateng [20] examine the impact of monetary policy on the risk-taking behaviour of Chinese banks in the presence of involuntary excess reserves which indicates unwanted surplus liquidity, yet they haven't examined the mechanism that monetary policy affects bank risk-taking through involuntary excess reserves or bank liquidity. Chen et al. [18] take bank liquidity as a control variable while examine the role of monetary policy and banks' riskiness. As bank liquidity is an important factor affecting bank risk, it is necessary to consider the mechanism of risk-taking channel operates through bank liquidity. It is therefore hypothesized that there is a transmission mechanism of monetary policy through bank liquidity channels, namely a liquidity risk-taking channel of monetary policy, that is, loose monetary policy reduces bank liquidity risk, and the reduction of liquidity risk encourages bank managers to engage in more active lending behaviour, which leads to the increase of bank risk.

Hypothesis 2. There is a transmission mechanism of risk-taking channel of monetary policy operating through bank liquidity.

## Data and variables

We use unbalanced bank-level panel data of 123 banks in China during the period of 2003–2018 obtained from Bureau van Dijk's Bankscope database and the annual reports of the sample banks. Only commercial banks are included in our sample (state-owned commercial banks, joint-stock commercial banks, city commercial banks and rural commercial banks). Policy banks, cooperative banks and investment banks are excluded because they have different objectives rather than profitability. Macro data (including national and economic and financial indicators are obtained from Wind Economic Database.

## Banks' risk-taking

Our main proxy for bank risk is the banks' z-score which is widely used in the literature [21–24]. The z-score is computed as follows:

$$Z_{i,t} = \frac{ROA_{i,t} + CAR_{i,t}}{\sigma_i(ROA_{i,t})} \tag{1}$$

where ROA is the return on assets, CAR is the capital asset ratio, and σ(ROA) is the standard deviation of ROA over a rolling window of 3-year period. The z-score measures the distance from insolvency in standard deviations. It represents the number of standard deviations that profits have to fall for the bank to become insolvent [22]. Following Laeven and Levine [21], we use the logarithmic transformation for z-score to deal with extreme values, denoted LnZ. A higher z-score value indicates lower probability of insolvency and greater bank stability. To facilitate interpretation, we denote "–LnZ" as the negative of the log transformed z-score values. We later use the change in non-performing loan ratio (dNPL) to measure banks' risk-taking when we conduct robustness tests.

## Bank liquidity

Bank liquidity has been measured in the literature in many different ways and there is little consensus from the theoretical literature on how to measure bank-specific funding liquidity risk. Drehmann and Nikolaou [7] define bank funding liquidity risk as the bank's failure to solve obligations immediately. Diamond and Dybvig [25] argue that banks with excessive deposits face lower liquidity risk in the presence of deposit insurance because depositors are unlikely to "run" when their deposits are protected. The theoretical research of Acharya and Naqvi [17] measure banks' funding liquidity risk by deposits because deposits shield banks from "run" risk and find that banks having higher deposits will take more risk. Khan et al. [1] argue that banks having higher deposits face lower liquidity risk because they have enough funds to solve their obligations and there is less "run" risk in the presence of deposit insurance. Although China only officially implemented the deposit insurance system in 2015, China has in fact implemented the hidden deposit insurance for a long time. Therefore, this paper refers to Khan et al. [1] and Acharya and Naqvi [17], and takes the ratio of deposits to total assets as the proxy variable of bank liquidity. In order to verify the robustness of the results, we also employ the negative of loans to deposits ratio (LTD) to measure bank liquidity. A higher value of deposits to total assets ratio (Liquidity) and LTD denote higher funding liquidity and face lower liquidity risk.

## Monetary policy

Interest rate, money supply and reserve requirement rate are the main instruments of monetary policy in China. Although China has basically liberalized the interest rate control in 2015, interest rate is yet to be fully marketized. Therefore, we use the 7-day interbank lending rate (Interbank) as interest rate indicators. As the reserve requirement rate is one of monetary policy instruments which has a direct impact on the volume of credit and thus the willingness of the financial system to take risks [26] and the M2 growth is managed as an intermediary monetary goal, we also use the reserve requirement rate (Reserve) and the negative of M2 growth rate (M2) to strengthen the robustness of our results. The reserve requirement ratio and the interbank lending rate are all weighted by time to obtain annual data. The higher value of the three monetary policy variables indicate the tighter monetary policy.

## The set of control variables

In order to avoid omitted-variable bias, we control for a series of bank-specific characteristics and general economic conditions that may also affect banks' risk-taking behaviour. We include five bank-specific characteristics that are commonly adopted in the literature.

Bank size(Size)-this variable is proxied by the natural logarithm of real total assets. Bank size is an important factor affecting banks' risk-taking behaviour. Banks will make risk decisions based on their size, and Khan et al. [1] suggest that larger banks will take less risk in response to lower funding liquidity risk. Considering the possible endogeneity of this variable, we refer to Khan et al. [1]and adopt its first lag value.

Loans to assets ratio (Loan)- Loan is measured as the ratio of loans to total assets. Some empirical studies find that banks that lend higher are generally riskier [1, 4]. Consistent with Khan et al. [1] and Rokhim and Min [4], we adopt its first lag value to deal with possible problems of endogeneity.

Bank capitalization (Capitalization)-this variable is measured as the ratio of capital to total assets. Some empirical studies provide evidence that higher capitalization effectively limit the risk-taking incentives of banks [9, 18], while Calem and Rob [27] suggest that banks may take more risk as their capitalization exceeds a certain threshold.

Bank efficiency (Efficiency)- Efficiency is represented by the cost income ratio. A higher value in our measure implies lower bank efficiency. Efficient banks may be more capable in managing risks, and thus bank riskiness declines [9].

National banks (National)-We also control for different types of banks by including National dummy which has the value of 1 for national banks and 0 for province banks.

In addition, we also include three macroeconomic variables into our control variables, namely, the growth rate of real GDP (GDP), the growth rate of house price (House) and Crisis. In consideration of the business area of different banks, national banks use country-level data of the three macroeconomic variables, while local banks use the province-level data. Crisis is a dummy variable which have a value of 1 in the year 2008 and 0 otherwise which control for the effect of Global Financial Crisis on Chinese banks.

## Descriptive statistics

Table 1 presents the definition of various variables and the characteristics of our sample, which consists of annual data of 123 banks in China during the period of 2003–2018. The negative of log transformed z-score has the mean of -4.86 and ranges from -8.12 to -0.25. The fairly high standard deviation and the wide range of Z-scores imply a considerable variation on the level of riskiness across banks. On average, bank liquidity, capitalization and efficiency for our sample constitute 73.4%, 6.7% and 38.9%. With regard to monetary policy, Reserve, Interbank and M2 constitute 16.6%, 3.0% and 14.7% on average.

## Model and methodology

Given the high persistence of bank risk-taking, this paper refers to the practice of Jimenez et al. [19] and introduces the first and second lag of risk variables as explanatory variables in the dynamic model. Our benchmark econometric model is described as follows:

$$Risk_{i,t} = \alpha_0 + \alpha_1 Risk_{i,t-1} + \alpha_2 Risk_{i,t-2} + \alpha_3 Liquidity_{i,t-1} + Controls_{i,t}\ \beta_C + \mu_i + \varepsilon_{i,t} \qquad (2)$$

Where $Risk_{i,t}$ is the dependent variable measured as the negative of z-score, indicating the risk for bank i in year t. Liquidity is the proxy variable of bank liquidity and Controls are a set

**Table 1. Descriptive statistics.**

| Variable | Description | Observations | Mean | Std.Dev. | Min. | Max. |
|---|---|---|---|---|---|---|
| -LnZ | Negative of log transformed z-score | 1276 | -4.860 | 0.966 | -8.117 | -0.253 |
| dNPL | The change in non-performing loan ratio | 1387 | 0.235 | 1.876 | -20.11 | 38.19 |
| Liquidity | Proxy by the ratio of deposits to total assets | 1562 | 0.734 | 0.116 | 0.395 | 1.179 |
| LTD | Negative of loans to deposits ratio | 1559 | -0.655 | 0.116 | -1.109 | -0.210 |
| Reserve | The reserve requirement rate (%) | 1571 | 16.55 | 3.926 | 6.330 | 20.83 |
| Interbank | The interbank lending rate (%) | 1544 | 3.024 | 0.804 | 1.280 | 4.190 |
| M2 | Negative of M2 growth rate (%) | 1571 | -14.70 | 4.889 | -27.68 | -8.1 |
| Size | Natural logarithm of assets | 1563 | 11.826 | 1.706 | 8.070 | 17.137 |
| Loan | The ratio of loans to total asset | 1556 | 0.479 | 0.104 | 0.145 | 0.775 |
| Capitalization | The ratio of equity to total assets | 1563 | 0.067 | 0.039 | -0.137 | 0.925 |
| Efficiency | The cost income ratio (%) | 1557 | 38.87 | 10.85 | 16.80 | 206.1 |
| National | Dummy = 1 if bank is national bank | 1571 | 0.165 | 0.372 | 0 | 1 |
| Crisis | Dummy = 1 in year 2008 | 1571 | 0.064 | 0.245 | 0 | 1 |
| GDP | Real GDP growth rate (%) | 1571 | 9.92 | 2.96 | -2.5 | 19.2 |
| House | The growth rate of house price (%) | 1571 | 10.38 | 7.86 | -13.75 | 52.36 |

of control variables. $\mu$ is the time-invariant bank-specific effect and $\varepsilon$ is the error term. $\alpha$ and $\beta$ are the coefficients to be estimated.

Following Khan et al. [1], we assume that there is a lagged relation between bank liquidity and bank risk-taking, i.e., a reduction in liquidity risk increases bank's risk-taking in the next period. Therefore, we introduce the first lag of bank liquidity as explanatory variable and examine whether a decrease in liquidity risk measured by more deposits will induce bank managers to take more risk. If $\alpha_3$ is significantly positive, it indicates that an increase in bank liquidity leads to an increase in bank risk.

In order to further investigate the relationship between bank liquidity, monney policy and risk-taking, we extend the baseline model by including a set of monetary policy variables and the interaction between bank liquidity and monetary policy variables:

$$Risk_{i,t} = \alpha_0 + \alpha_1 Risk_{i,t-1} + \alpha_2 Risk_{i,t-2} + \alpha_3 Liquidity_{i,t-1} + \alpha_4 MP_{i,t} + \alpha_5 MP_{i,t} \times Liquidity_{i,t-1}$$
$$+ Controls_{i,t}\ \beta_C + \mu_i + \varepsilon_{i,t} \qquad (3)$$

This paper focuses on the sign and significance of $\alpha_4$ and $\alpha_5$. If $\alpha_4$ is significantly negative, it indicates that loose monetary policy encourages banks to take more risk and the risk-taking channel of monetary policy exists. If $\alpha_5$ is significant, it indicates that the impact of monetary policy on bank risk-taking depends on the bank liquidity.

Finally, this paper uses the mediation model to further test the existence of a liquidity risk-taking channel of monetary policy. Following Baron and Kenny's [28] causal steps approach, we build the following models:

$$Risk_{i,t} = \alpha_0 + \alpha_1 Risk_{i,t-1} + \alpha_2 Risk_{i,t-2} + \alpha_3 MP_{i,t} + Controls_{i,t}\ \beta_C + \mu_i + \varepsilon_{i,t} \qquad (4)$$

$$Liquidity_{i,t} = \beta_0 + \beta_1 Liquidity_{i,t-1} + \beta_1 Liquidity_{i,t-2} + \beta_3 MP_{i,t} + Controls_{i,t}\ \pi_C + \mu_i + \varepsilon_{i,t} \quad (5)$$

$$Risk_{i,t} = \lambda_0 + \lambda_1 Risk_{i,t-1} + \lambda_2 Risk_{i,t-2} + \lambda_3 Liquidity_{i,t-1} + \lambda_4 MP_{i,t} + Controls_{i,t} \nu_C + \mu_i + \varepsilon_{i,t} \quad (6)$$

Model (4) is the first step to test the total effect of monetary policy on bank risk-taking. If $\alpha_3$ is significantly negative, it indicates the existence of bank risk-taking channels of monetary

policy, and its estimated coefficient is the total effect of monetary policy on bank risk-taking. Model (5) is the second step to test whether monetary policy has effect on the intermediary variable (bank liquidity). If $\beta_3$ significantly negative, it indicates that loose monetary policy will reduce bank liquidity risk. Model (6) is the third step to test whether the intermediary variable is the complete intermediary or partial intermediary of monetary policy affecting bank risk-taking. If $\lambda_3$ is significant and $\lambda_4$ is not significant, it indicates a complete mediation. If $\lambda_3$ and $\lambda_4$ are both significant, and $\lambda_4$ is lower than $\alpha_3$, it indicates that there is a partial mediation and the mediating effect accounts for $\beta_3 \times \lambda_3 / \alpha_3$ of the total effect.

## Results

As the empirical models in this paper are dynamic panel models (where containing the lagged dependent variable), the ordinary least squares estimator (OLS) and fixed effect estimators are biased. To solve this problem, Arellano and Bond [29] propose the generalized method of moments (GMM) estimators, which include difference GMM estimators and system GMM estimators. According to Arellano and Bover [30] and Blundell and Bond [31], system GMM estimator augments difference GMM estimator by estimating simultaneously in differences and levels, which makes up for the deficiency of difference GMM estimator and enhances the validity of instruments in difference equation. Therefore, system GMM estimator is used to estimate the models in this paper. System GMM is implemented by command xtabond2 on STATA package. The bank-level explanatory variables are considered as predetermined, while macroeconomic and financial variables are treated as strictly exogenous variables.

### Bank liquidity and risk-taking

To investigate the impact of bank liquidity on bank risk-taking, we first estimate the model (2), and the results are shown in Table 2. The system GMM estimators include one-step and two-step estimators. However, the one-step GMM estimator will be used since it has been shown to result in more reliable inferences. The asymptotic standards errors from the two step GMM estimator have been found to have a downward bias [31]. Therefore, although we compare the one-step and two-step estimators in Table 2, we will only use the one-step estimator in the subsequent estimation.

As shown in Table 2, column (1) and (2), bank liquidity has a positive and significant impact on bank risk, indicating that lower liquidity risk encourage banks to take more risks which is supportive of our first hypothesis. The results are similar with the empirical study of Khan et al. [1] on U.S. Bank Holding Company data and are also consistent with the theoretical prediction of Acharya and Naqvi [17]. As shown in column (1), the impact of bank liquidity on bank risk is significant as a one standard deviation increase in a bank's deposit to total asset increases the natural logarithm of Z-score by 1.7. In order to test the robustness of the estimated results, the column (3) and (4) further introduce macroeconomic variables, and the coefficient of bank liquidity are still significant.

With regard to control variables, the coefficients of bank size are negative and significant indicating that bank size reduces the riskiness of banks, which is consistent with Delis and Kouretas [9] and Khan et al. [1]. The ratio of capital to total assets is negatively related to bank risk which suggests that higher capitalization effectively limit the risk-taking incentives of banks. The coefficients of the bank efficiency are positive and significant indicating that the increase of bank's cost income ratio will lead to the rise of bank risk. In addition, the coefficients of loans to total assets ratio, national bank dummy, real GDP growth and real estate price growth are not significant. Finally, turning to the test statistics, neither the Hansen test

**Table 2. The impact of bank liquidity on risk-taking.**

| Dependent variable: the Z-score (-LnZ) | | | | |
|---|---|---|---|---|
| | **(1) one-step** | **(2) two-step** | **(3) one-step** | **(4) two-step** |
| L.(-LnZ) | 0.520*** (9.50) | 0.517*** (9.39) | 0.525*** (9.76) | 0.514*** (9.16) |
| L2. (-LnZ) | -0.204*** (-5.73) | -0.214*** (-5.97) | -0.204*** (-5.73) | -0.223*** (-5.85) |
| L. Liquidity | 1.665*** (3.15) | 1.479** (2.55) | 1.528*** (3.00) | 1.283** (2.18) |
| L. Size | -0.136*** (-2.94) | -0.154*** (-3.13) | -0.112** (-2.44) | -0.130** (-2.53) |
| Loan | -0.704 (-1.66) | -0.623 (-1.36) | -0.616 (-1.46) | -0.526 (-1.07) |
| Capitalization | -2.315* (-1.77) | -2.245 (-1.56) | -2.457* (-1.79) | -2.429 (-1.41) |
| Efficiency | 0.012** (2.27) | 0.013** (2.25) | 0.013** (2.51) | 0.012* (1.90) |
| National | 0.239 (1.49) | 0.266 (1.48) | 0.165 (1.04) | 0.173 (0.95) |
| Crisis | | | -0.056 (-0.42) | -0.038 (-0.30) |
| GDP | | | 0.010 (0.81) | 0.014 (1.10) |
| House | | | 0.003 (0.90) | 0.004 (1.05) |
| Constant | -3.034*** (-4.60) | -2.807*** (-3.89) | -3.372*** (-4.48) | -3.179*** (-3.77) |
| Observations | 1,022 | 1,022 | 1,022 | 1,022 |
| AR(1) | 0.000 | 0.000 | 0.000 | 0.000 |
| AR(2) | 0.329 | 0.348 | 0.318 | 0.419 |
| Hansen (P) | 1.000 | 1.000 | 1.000 | 1.000 |

This table reports the system GMM results of the impact of bank liquidity on risk-taking for Chinese banks in the sample period of 2003–2018. All variables are defined in Table 1. We compare the one-step and two-step estimators of system GMM in columns (1)–(4). In column (1) and (2), we control bank-specific characteristics, while we further introduce macroeconomic variables in column (3) and (4). Numbers in brackets report the t-statistic. All tests are based on robust system GMM. The Hansen test p value is the test for overidentification. First order and Second order tests are test statistics for first and second order autocorrelations in residuals, respectively, under the null hypothesis of no serial correlation.

***, **, * indicate coefficient significance at the 1%, 5% and 10% levels, respectively.

nor the second order statistics provide evidence that rejects the validity of the instruments and the no serial correlation assumption.

## The liquidity risk-taking channel of monetary policy

In order to examine whether the relationship between bank liquidity and bank risk is affected by monetary policy, we first estimate the model (3) without the interaction terms between bank liquidity and monetary policy. The results are reported in column (1) to column (3) in Table 3. Then, we introduce the interaction terms to further investigate the relationship between bank liquidity, monetary policy and bank risk, and the results are shown in column (4) to column (6) in Table 3.

As shown in column (1) to column (3), the estimated coefficients of monetary policy proxy variables are all significant, in which the reserve requirement rate, the interbank lending rate and the negative of M2 growth rate are all significantly negative. This means that loose monetary policy will cause banks to take more risks, which provides additional evidence for a risk-taking channel of monetary policy in China. At the same time, the coefficients of bank liquidity variables are significantly positive, which indicates that after further controlling the monetary policy variables that affect market liquidity, the improvement of bank liquidity still leads to higher bank risk-taking.

As shown in column (4) to column (6), except the the interaction terms between M2 and bank liquidity, the other interaction terms are significantly negative, indicating that the impact of monetary policy on bank risk-taking depends on the bank's liquidity risk, and banks with

**Table 3. The impact of bank liquidity and monetary policy on risk-taking.**

| Dependent variable: the Z-score (-LnZ) | | | | | | |
|---|---|---|---|---|---|---|
| | model (3) without the interaction terms | | | model (3) | | |
| | **(1) Reserve** | **(2) Interbank** | **(3) M2** | **(4) Reserve** | **(5) Interbank** | **(6) M2** |
| L.(-LnZ) | 0.500*** (9.25) | 0.521*** (9.65) | 0.534*** (10.00) | 0.463*** (8.95) | 0.516*** (9.42) | 0.569*** (7.57) |
| L2.(-LnZ) | -0.220*** (-6.14) | -0.217*** (-5.68) | -0.220*** (-5.83) | -0.233*** (-5.45) | -0.213*** (-4.78) | -0.264*** (-3.65) |
| MP | -0.031*** (-2.84) | -0.060* (-1.90) | -0.026*** (-3.23) | 0.201** (2.47) | 0.333 (1.43) | -0.104** (-2.02) |
| L. Liquidity | 1.811*** (3.34) | 1.664*** (3.29) | 1.334*** (2.72) | 6.150*** (3.26) | 2.529** (2.50) | 1.869* (1.88) |
| MP×L. Liquidity | | | | -0.295*** (-2.87) | -0.503* (-1.73) | 0.099 (1.50) |
| L. Size | -0.118** (-2.53) | -0.109** (-2.35) | -0.074 (-1.64) | -0.182*** (-3.66) | -0.134*** (-2.69) | -0.112 (-1.60) |
| L.Loan | -0.910** (-2.02) | -0.763* (-1.78) | -0.679* (-1.69) | -0.287 (-0.62) | -0.214 (-0.55) | -0.590 (-1.21) |
| Capitalization | -1.998 (-1.56) | -2.214* (-1.71) | -2.037 (-1.58) | -1.746*** (-4.09) | -2.098* (-1.84) | -1.777*** (-2.93) |
| Efficiency | 0.014** (2.57) | 0.013** (2.48) | 0.010* (1.77) | 0.013** (2.25) | 0.006 (1.17) | 0.011* (1.76) |
| National | 0.173 (1.08) | 0.158 (0.98) | 0.008 (0.05) | 0.286 (1.65) | 0.189 (1.13) | 0.092 (0.44) |
| Crisis | -0.077 (-0.58) | -0.060 (-0.45) | -0.107 (-0.77) | -0.048 (-0.38) | -0.074 (-0.57) | -0.101 (-0.79) |
| GDP | 0.006 (0.52) | 0.011 (0.87) | -0.002 (-0.12) | 0.003 (0.20) | 0.008 (0.64) | 0.007 (0.51) |
| House | 0.000 (0.06) | 0.001 (0.14) | -0.002 (-0.37) | 0.001 (0.35) | 0.000 (0.04) | -0.000 (-0.06) |
| Constant | -2.996*** (-3.97) | -3.332*** (-4.47) | -3.784*** (-5.41) | -6.242*** (-3.64) | -3.718*** (-3.57) | -4.029*** (-3.65) |
| Observations | 1,022 | 1,022 | 1,022 | 1,022 | 1,022 | 1,022 |
| AR(1) | 0.000 | 0.000 | 0.000 | 0.000 | 0.000 | 0.000 |
| AR(2) | 0.415 | 0.448 | 0.484 | 0.507 | 0.465 | 0.813 |
| Hansen P | 1.000 | 1.000 | 1.000 | 1.000 | 1.000 | 0.997 |

This table reports the one-step system GMM results of the impact of monetary policy and bank liquidity on risk-taking for Chinese banks in the sample period of 2003–2018. All variables are defined in Table 1. We introduce three monetary policy proxy variables in model (2) and model (3), respectively. Numbers in brackets report the t-statistic. All tests are based on robust system GMM. The Hansen test p value is the test for overidentification. First order and Second order tests are test statistics for first and second order autocorrelations in residuals, respectively, under the null hypothesis of no serial correlation. ***, **, * indicate coefficient significance at the 1%, 5% and 10% levels, respectively.

higher liquidity risk are more sensitive to loose monetary policy. The impact of the monetary policy on risk-taking is shown as follows:

$$\frac{\partial \text{Risk}}{\partial \text{MP}} = \alpha_4 + \alpha_5 \times \text{Liquidity} \tag{7}$$

As $\alpha_5$ is negative and significant in column (4) and column (5), banks with higher liquidity tend to increase risk-taking more rapidly in response to the loose monetary policy. Similarly, Borio and Zhu [6] believes that the self-reinforcing link between liquidity and risk-taking could potentially have a material effect on the strength of the transmission of monetary policy impulses, akin to a "multiplier" effect. Higher liquidity weakens spending constraints of banks, while weaker constraints will allow bank managers to lend more aggressively and ultimately increase the riskiness of banks. This means that while the long-term loose monetary policy reduces the liquidity risk of banks, it may stimulate the risk-taking behavior of banks. Therefore, we can preliminarily infer that the liquidity risk-taking channel of monetary policy exists in China which is supportive of our second hypothesis.

To further test the existence of a liquidity risk-taking channel of monetary policy, we estimate the mediation model, and the results are shown in Table 4. Firstly, as shown in column (1) to column (3), the estimations of monetary policy in model (4) are all significantly negative, indicating that loose monetary policy will increase bank risk-taking and the total effect of

**Table 4. The liquidity risk-taking channel of monetary policy–mediation model.**

| | Dependent variable | | | | | | | | |
|---|---|---|---|---|---|---|---|---|---|
| | -LnZ in model (4) | | | Liquidity in model (5) | | | -LnZ in model (6) | | |
| | (1) Reserve | (2) Interbank | (3) M2 | (4) Reserve | (5) Interbank | (6) M2 | (7) Reserve | (8) Interbank | (9) M2 |
| L. LnZ | 0.380*** (6.55) | 0.413*** (7.14) | 0.536*** (11.45) | | | | 0.500*** (10.31) | 0.521*** (9.65) | 0.534*** (10.00) |
| L2. LnZ | -0.239*** (-5.05) | -0.227*** (-4.43) | -0.186*** (-4.61) | | | | -0.200*** (-5.66) | -0.217*** (-5.68) | -0.220*** (-5.83) |
| L. Liquidity | | | | 0.776*** (11.61) | 0.739*** (17.72) | 0.727*** (11.05) | 1.465*** (2.94) | 1.664*** (3.29) | 1.334*** (2.72) |
| L2. Liquidity | | | | -0.022 (-0.44) | -0.031 (-0.94) | -0.048 (-0.96) | | | |
| MP | -0.034*** (-2.88) | -0.059* (-1.80) | -0.032*** (-4.44) | -0.002*** (-2.70) | -0.003 (-1.43) | -0.002*** (-4.21) | -0.027** (-2.54) | -0.060* (-1.90) | -0.026*** (-3.23) |
| L. Size | -0.224*** (-3.45) | -0.201*** (-3.16) | -0.098** (-1.99) | -0.001 (-0.47) | -0.003 (-0.92) | 0.001 (0.33) | -0.124*** (-2.66) | -0.109** (-2.35) | -0.074 (-1.64) |
| Loan | 0.234 (0.60) | 0.303 (0.79) | 0.266 (0.91) | 0.074* (1.70) | 0.115*** (4.04) | 0.118*** (2.80) | -0.664 (-1.59) | -0.763* (-1.78) | -0.679* (-1.69) |
| Capitaliza-tion | -1.831*** (-4.80) | -1.932*** (-4.37) | -1.066 (-0.91) | -0.003 (-0.10) | 0.006 (0.14) | -0.011 (-0.39) | -2.178* (-1.87) | -2.214* (-1.71) | -2.037 (-1.58) |
| Efficiency | 0.018*** (3.11) | 0.019*** (3.38) | 0.004 (0.88) | 0.001* (1.67) | 0.001** (2.32) | 0.000 (1.14) | 0.012** (2.51) | 0.013** (2.48) | 0.010* (1.77) |
| National | 0.361* (1.66) | 0.322 (1.52) | 0.022 (0.13) | -0.011 (-1.10) | -0.008 (-0.86) | -0.024* (-1.92) | 0.180 (1.11) | 0.158 (0.98) | 0.008 (0.05) |
| Crisis | -0.054 (-0.43) | -0.033 (-0.26) | -0.189 (-1.44) | 0.033*** (4.23) | 0.034*** (4.90) | 0.032*** (4.39) | -0.099 (-0.78) | -0.060 (-0.45) | -0.107 (-0.77) |
| GDP | 0.019 (1.23) | 0.024 (1.60) | -0.000 (-0.04) | 0.001 (1.37) | 0.001* (1.79) | 0.001 (1.01) | 0.007 (0.54) | 0.011 (0.87) | -0.002 (-0.12) |
| House | 0.000 (0.04) | 0.001 (0.21) | -0.004 (-1.04) | 0.001** (2.39) | 0.001*** (3.14) | 0.001** (2.51) | 0.000 (0.01) | 0.001 (0.14) | -0.002 (-0.37) |
| Constant | -1.902** (-2.14) | -2.481*** (-2.96) | -2.703*** (-3.99) | 0.137** (2.49) | 0.148*** (2.83) | 0.093 (1.62) | -2.689*** (-3.75) | -3.332*** (-4.47) | -3.784*** (-5.41) |
| AR(2) | 0.510 | 0.505 | 0.264 | 0.981 | 0.842 | 0.639 | 0.299 | 0.448 | 0.484 |
| Hansen P | 1.000 | 1.000 | 1.000 | 0.983 | 1.000 | 1.000 | 1.000 | 1.000 | 1.000 |

This table reports the one-step system GMM results of mediation model. All variables are defined in Table 1. In column (1)—(3), we test the total effect of monetary policy on bank risk-taking. In column (4)—(6), we test whether monetary policy has effect on the banks liquidity risk. In column (7)—(9), we test whether the intermediary variable is the complete intermediary or partial intermediary. Numbers in brackets report the t-statistic. ***, **, * indicate coefficient significance at the 1%, 5% and 10% levels, respectively.

monetary policy on bank risk-taking are -0.031, -0.06 and -0.026, respectively. Secondly, as shown in column (4) to column (6), except the interbank lending rate, the estimations of monetary policy in model (5) are significantly negative, indicating that loose monetary policy will reduce bank liquidity risk. Finally, the coefficients of monetary policy and intermediary variable in column (7) and (9) are significant, and the coefficients of monetary policy are lower than the corresponding values in regression Eqs (1) and (3), indicating that there is a partial intermediary effect when monetary policy is proxied by the reserve requirement rate and the M2 growth rate. In the model with the reserve requirement rate as the proxy variable of monetary policy, the mediating effect accounts for 8.6% of the total effect, and the Sobel test value is -2.12, which is significant at the 5% confidence level. Moreover, in the model with the M2 growth rate as the proxy variable of monetary policy, the mediating effect accounts for 8.3% of the total effect, and the Sobel test value is -2.23, which is significant at the 5% confidence level. To sum up, the estimation results support our second hypothesis that the liquidity risk-taking channel of monetary policy exists in China.

## Robustness tests

To bolster the robustness of the findings previously obtained, we conduct various robustness tests on the benchmark model and the mediation model, and the estimated results are shown in Tables 5 and 6.

First, we use an alternative measure of bank liquidity- the negative of loans to deposits ratio (LTD). As shown in column (1)- (4) in Table 5, the coefficients of LTD are significantly positive in most regression equations and interaction terms between monetary policy and bank liquidity are significant in column (2) and column (4).

Second, we replace Z-scores by using the change in non-performing loan ratio to measure banks' risk-taking. As shown in column (5)—(8) in Table 5, the coefficients of bank liquidity are significantly positive in column (5), and interaction terms are significant in column (6) and column (8). Therefore, the estimation results in Table 5 also support the hypothesis that

**Table 5. Robustness tests.**

| | Alternative measure of bank liquidity (LTD) | | | | Alternative measure of bank risk-taking (dNPL) | | | |
|---|---|---|---|---|---|---|---|---|
| | model (2) | model (3) | | | model (2) | model (3) | | |
| | (1) | (2) Reserve | (3) Interbank | (4) M2 | (5) | (6) Reserve | (7) Interbank | (8) M2 |
| L. LnZ | 0.537*** (10.70) | 0.582*** (5.54) | 0.465*** (6.20) | 0.502*** (7.19) | | | | |
| L2. LnZ | -0.199*** (-5.23) | -0.482*** (-5.35) | -0.221*** (-3.45) | -0.213*** (-3.44) | | | | |
| L.dNPL | | | | | -0.167 (-1.53) | -0.118* (-1.82) | -0.053 (-1.28) | -0.215* (-1.83) |
| L2.dNPL | | | | | -0.104 (-1.49) | -0.031 (-0.73) | -0.037 (-1.09) | -0.108 (-1.63) |
| L.Liquidity/ L.LTD | 1.301** (2.31) | 6.702* (1.85) | -0.333 (-0.24) | 2.943** (2.45) | 4.806* (1.97) | -3.606 (-0.75) | 2.328 (0.73) | -5.514** (-2.04) |
| MP | | -0.241* (-1.86) | 0.111 (0.43) | 0.074 (1.43) | | -0.375* (-1.93) | -0.264 (-0.61) | 0.653** (2.39) |
| MP×L.Liquidity/ MP×L.LTD | | -0.325* (-1.67) | 0.259 (0.67) | 0.155* (1.95) | | 0.458* (1.71) | 0.269 (0.41) | -0.875** (-2.50) |
| L. Size | -0.137** (-2.25) | -0.333*** (-3.35) | -0.189** (-2.61) | -0.101 (-1.49) | 0.340* (1.82) | 0.151 (1.22) | 0.167 (1.31) | 0.158 (0.75) |
| L.Loan | 1.063** (1.99) | 0.934 (0.93) | 0.799 (1.31) | 0.320 (0.53) | -1.635 (-0.95) | -2.712 (-1.26) | -1.923 (-1.24) | -4.106** (-1.98) |
| Capitalization | -1.770*** (-3.41) | -3.848 (-1.29) | -1.808*** (-3.95) | -1.151** (-2.20) | -0.290 (-0.29) | -0.226 (-0.07) | 0.862 (0.29) | -0.049 (-0.04) |
| Efficiency | 0.012* (1.97) | 0.012 (1.49) | 0.015** (2.28) | 0.014** (2.21) | 0.011 (0.73) | 0.021** (2.06) | 0.013 (1.43) | 0.012 (0.56) |
| National | 0.262 (1.41) | 0.705** (2.22) | 0.331 (1.47) | 0.096 (0.46) | -0.593 (-1.35) | 0.025 (0.09) | -0.130 (-0.52) | 0.165 (0.33) |
| Crisis | -0.077 (-0.63) | -0.048 (-0.35) | -0.078 (-0.61) | -0.101 (-0.77) | 1.516** (2.26) | 1.203* (1.89) | 1.253** (1.98) | 1.404** (2.21) |
| GDP | 0.017 (1.18) | 0.006 (0.28) | 0.017 (1.19) | 0.011 (0.78) | 0.103*** (2.75) | 0.078*** (3.65) | 0.080*** (4.61) | 0.076*** (2.98) |
| House | 0.002 (0.62) | 0.003 (0.57) | 0.000 (0.02) | -0.002 (-0.48) | 0.025*** (4.49) | 0.017*** (3.11) | 0.020*** (2.94) | 0.005 (0.85) |
| Constant | -1.910*** (-2.84) | 3.637 (1.24) | -2.577* (-1.93) | -1.538* (-1.85) | -8.172** (-2.21) | 1.230 (0.34) | -3.957 (-1.22) | 2.921 (1.11) |
| AR(2) | 0.226 | 0.208 | 0.531 | 0.479 | 0.477 | 0.987 | 0.412 | 0.558 |
| Hansen P | 1.000 | 1.000 | 1.000 | 1.000 | 0.923 | 1.000 | 1.000 | 1.000 |

This table reports the estimation results of various robustness tests by using alternative alternative measure of bank liquidity and bank risk-taking. All variables are defined in Table 1. In column (1) to (4), we replace the ratio of deposits to total assets by using the negative of loans to deposits ratio to measure bank liquidity. In column (5) to (8), we replace the Z-scores by using the change in non-performing loan ratio to measure banks' risk-taking. Numbers in brackets report the t-statistic. ***, **, * indicate coefficient significance at the 1%, 5% and 10% levels, respectively.

**Table 6. Robustness tests–mediation model.**

| | Dependent variable | | | | | |
|---|---|---|---|---|---|---|
| | Liquidity in model (5) | | | -LnZ in model (6) | | |
| | (1) Reserve | (2) Interbank | (3) M2 | (4) Reserve | (5) Interbank | (6) M2 |
| L.LTD | 0.908*** (15.68) | 0.952*** (21.32) | 0.908*** (17.73) | 1.211*** (2.88) | 1.345*** (2.76) | 1.319** (2.31) |
| L2.LTD | -0.134*** (-3.53) | -0.117*** (-3.58) | -0.131*** (-3.41) | | | |
| L.LnZ | | | | 0.534*** (11.87) | 0.524*** (11.05) | 0.492*** (9.13) |
| L2. LnZ | | | | -0.187*** (-4.97) | -0.176*** (-4.40) | -0.196*** (-4.43) |
| MP | 0.004*** (4.73) | -0.004* (-1.94) | -0.002*** (-4.93) | -0.026*** (-2.62) | -0.052* (-1.69) | -0.027*** (-3.50) |
| L. Size | -0.004 (-1.06) | -0.002 (-0.44) | -0.001 (-0.25) | -0.151*** (-3.38) | -0.141*** (-2.64) | -0.107* (-1.89) |
| L.Loan | 0.022 (0.44) | 0.015 (0.33) | -0.008 (-0.17) | 1.237*** (2.89) | 1.287** (2.55) | 1.051* (1.80) |
| Capitalization | -0.150** (-2.21) | -0.101** (-2.37) | -0.107** (-2.27) | -1.862** (-2.06) | -2.163* (-1.87) | -1.890 (-1.57) |
| Efficiency | 0.001*** (3.44) | 0.001*** (3.20) | 0.001** (2.17) | 0.004 (0.80) | 0.006 (1.22) | 0.004 (0.78) |
| National | -0.010 (-0.97) | -0.017 (-1.61) | -0.026** (-2.33) | 0.265* (1.73) | 0.257 (1.50) | 0.109 (0.59) |
| Crisis | 0.014 (1.38) | 0.011 (1.06) | 0.005 (0.48) | -0.164 (-1.30) | -0.124 (-0.98) | -0.113 (-0.86) |
| GDP | 0.005*** (4.13) | 0.005*** (4.55) | 0.003*** (2.77) | 0.002 (0.14) | 0.007 (0.50) | -0.004 (-0.26) |
| House | 0.000 (0.51) | -0.001** (-2.50) | -0.001*** (-4.05) | -0.001 (-0.30) | -0.001 (-0.15) | -0.003 (-0.60) |
| Constant | -0.256*** (-4.34) | -0.158*** (-3.02) | -0.206*** (-3.63) | -0.878 (-1.23) | -1.347* (-1.96) | -2.277*** (-3.01) |
| AR(2) | 0.140 | 0.120 | 0.224 | 0.229 | 0.216 | 0.372 |
| Hansen P | 1.000 | 1.000 | 1.000 | 1.000 | 1.000 | 1.000 |

This table reports the robustness tests of mediation model by adopt the negative of loans to deposits ratio (LTD) as the proxy for bank liquidity. All variables are defined in Table 1. In column (1)—(3), we test whether monetary policy has effect on the banks liquidity risk. In column (4)—(6), we test whether the intermediary variable is the complete intermediary or partial intermediary. Numbers in brackets report the t-statistic. ***, **, * indicate coefficient significance at the 1%, 5% and 10% levels, respectively.

the risk-taking of Chinese banks has a positive relationship with bank liquidity and the impact of monetary policy on bank risk-taking depends on the bank's liquidity risk.

Finally, to further ascertain the robustness of our results about the liquidity risk-taking channel of monetary policy, we re-estimate the mediation model by adopt the negative of loans to deposits ratio (LTD) as the proxy for bank liquidity. As the estimations of model (4) is the same as the results of column (1)—(3) in Table 5, we just present the estimations of model (5) and model (6) in Table 6. As shown in Table 6, the estimated coefficients on monetary policy and bank liquidity are mostly consistent with the previous findings and the liquidity risk-taking channel of monetary policy exists when monetary policy is proxied by the interbank lending rate and the M2 growth rate. To sum up, the estimation results of this paper are relatively robust.

## Conclusions

In this paper, we have attempted to conduct empirical research to investigate the effect of banks' liquidity risk on risk-taking behaviour of Chinese banks, and provide evidence for a risk-taking channel of monetary policy operating through bank liquidity. We use panel data set for 123 Chinese commercial banks over the period of 2003–2018 and use the system GMM method to account for the possibility of dynamics and endogeneity issues. Our empirical results indicate that banks facing lower liquidity risk will be encouraged to take more risk. Moreover, loose monetary policy leads to more aggressive risk-taking by reducing the bank liquidity risk, namely a liquidity risk-taking channel of monetary policy, i.e., loose monetary policy reduces bank liquidity risk and the decrease in liquidity risk encourages banks to take more risk.

As far as policy implications are concerned, this study suggests that facing loose monetary policy and low liquidity risk, bank executives should still give full consideration to credit risk and adopt prudent lending policies because of the existence of a liquidity risk-taking channel of monetary policy. At the same time, as banks with higher liquidity risk are more sensitive to loose monetary policy, the regulatory authorities should take into account the influence of monetary policy on bank risk-taking through bank liquidity channels when monetary policy is deliberated, and comprehensively design the regulatory framework to effectively restrain the reverse incentive of changes in bank deposits to bank risks, and strengthen the supervision of highly liquid banks.

## Supporting information

**S1 Data.**
(ZIP)

## Author Contributions

**Writing – original draft:** Lihuan Zhuang.

**Writing – review & editing:** Cong Wang.

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
