## [Decision Letter · Decision Letter 0]

29 Aug 2022

PONE-D-22-18762Bank liquidity and the risk-taking channel of monetary policy : An empirical study of the banking system in ChinaPLOS ONE

Dear Dr. Zhuang,

Thank you for submitting your manuscript to PLOS ONE. After careful consideration, we feel that it has merit but does not fully meet PLOS ONE’s publication criteria as it currently stands. Therefore, we invite you to submit a revised version of the manuscript that addresses the points raised during the review process.

For the paper to be accepted all econometric work should be corrected along the lines proposed by both referees. 

We look forward to receiving your revised manuscript.

Kind regards,

Daphne Nicolitsas

Academic Editor

PLOS ONE

Journal Requirements:

“This work was supported by Chinese National Funding of Social Sciences (18BJL116).”

“YES - This work was supported by Chinese National Funding of Social Sciences (18BJL116).”

“YES - This work was supported by Chinese National Funding of Social Sciences (18BJL116).”

Reviewers' comments:

Reviewer's Responses to Questions

**Comments to the Author**

1. Is the manuscript technically sound, and do the data support the conclusions?

Reviewer #1: Partly

Reviewer #2: Partly

2. Has the statistical analysis been performed appropriately and rigorously? 

Reviewer #1: Yes

Reviewer #2: No

3. Have the authors made all data underlying the findings in their manuscript fully available?

Reviewer #1: Yes

Reviewer #2: No

4. Is the manuscript presented in an intelligible fashion and written in standard English?

Reviewer #1: Yes

Reviewer #2: Yes

5. Review Comments to the Author

Reviewer #1: The manuscript examines the impact of bank liquidity on risk-taking behavior of Chinese banks, and provides evidence for a risk-taking channel of monetary policy operating through bank liquidity. The author(s) use bank-level panel data for 123 Chinese commercial banks during the period of 2003-2018, and find that banks facing lower liquidity risk are encouraged to take more risk. The author(s) demonstrate a good motivation for undertaking this research and appropriately demonstrate their contribution to the existing literature on risk-taking channel of monetary policy transmission. Yet, I have some concerns and recommendations as listed below:

1. The novelty of their work seems to be related to the application of the concept of liquidity risk-taking channel of monetary policy. However, this concept has been used in the banking literature for a long time (see Acharya and Naqvi(2012), Khan et al.(2017), Borio and Zhu (2012), Chen et al (2017) etc.). Therefore, the author(s) have to clearly differentiate from other similar studies in the same area (e.g., Khan et al.(2017) and Acharya and Naqvi(2012)) and better define their contribution related specifically to the Chinees banking market. Also, it is not clear if the findings can be generalized to other developing economies.

2. In the same context, there is a lack of justification of why they undertake this research (e.g., why China banking is worth to explore and how the Chinese banking market is different from other developing markets). The authors could give more interest to their research by presenting in the introduction some stylized facts (with some graphs and/or tables) about the Chinese banking sector and its performance during the observation period (2003 - 2018). Why the recent global crisis (COVID-19) effect is not included in the analysis?

3. The hypotheses are extracted from an analysis of the existing studies without analyzing the existing gaps and puzzling issues that remain unsolved. Based on this unsolved issue(s) the authors should express their own expectations. It is well known fact that the presentation of the literature should be thorough, analytical, and able to provide both positive and negative side of existing theoretical approaches. So, gap analysis should be strengthened.

4. The choice of a proxy of the liquidity risk is not well justified. On page 3, the author(s) said: Following Acharya and Naqvi(2012), Khan et al.(2017) use the ratio of total deposits to total assets as the proxy for funding liquidity risk and find that a reduction in funding liquidity risk increases bank risk by using data for U.S. bank holding companies. However, using total deposits to total assets ratio as a measure of bank liquidity risk is not the most appropriate choice. Many studies use other measures such as Liquid Assets as percentage of Total Assets (see Mirzae et al. (2020, AOR), and Mateev et al. (2022, RIBF)) or Loans to Assets ratio (see González et al (2017, RIBF), and Louati et al. (2015, BIR). Therefore, the authors need to run robustness checks using some alternative measures of liquidity risk.

5. In general, the methodology is appropriate and rigorous. Thought the inclusion of the interaction term in the model is widely used technique, the analysis of the estimated coefficients of the interaction term should be done with a high degree of caution. It is required the sign and the magnitude of the variables taken as stand-alone variables and those of the interaction term itself to be taken concurrently. Also, it is recommended to include other bank-specific characteristics that are found to be relevant determinants of bank stability (e.g., Net Loan/Assets, Other earning assets, Income diversity, Tier 1, Liquid assets, etc.). Also, the authors should control for time differences using year dummies and the crisis effect through 2017-2018 Crisis dummy indicator. Differences in liquidity (and risk) between different types of banks (national banks vs. province banks) should also be controlled in the regressions.

6. Next, the purpose of using the mediation model to further test the liquidity risk-taking channel of monetary policy is not well justified. More specifically, the evidence indicating that there is a partial intermediary effect is not convincing. Therefore, a deeper explanation of the model and the outcomes of the results in Table 5 is required.

7. Finally, the study does not provide evidence how their findings will assist bank managers in evaluating the impact of liquidity as risk-taking channel of monetary policy. Policy implications for regulators and decision-makers proposing reforms/initiatives that are directed to the impact of monetary policy on bank risk-taking should be better formulated.

Reviewer #2: Referee Report for PONE-D-22-18762

“Bank liquidity and the risk-taking channel of monetary policy :

An empirical study of the banking system in China”

Overview

This paper investigates how bank liquidity and monetary policy are associated with banks’ risk-taking behaviour using a sample of 123 commercial banks in China. The authors argue that banks are more likely to take risk when they are subjected to lower liquidity risk and loose monetary policy. In addition, the study shows the incremental effect of bank liquidity on the risk-taking channel of monetary policy.

The paper is well written and well-structured. But I have some serious concerns regarding the paper’s contributions and empirical results. Below I state my detailed comments.

Research findings and contributions

The paper considers the impact of bank liquidity and monetary policy on bank risk taking. However, both leakages have already been examined in the extant literature on banks’ risk-taking behaviour. In addition, many prior studies have documented similar findings: (1) banks with higher liquidity take more risk (e.g., Acharya and Naqvi, 2012; Khan, Scheule, and Wu, 2017; Rokhim and Min, 2020); (2) loose monetary policy causes banks to take more risk (e.g., Angeloni, Faia, and Duca, 2015; Chen, Wu, Jeon, and Wang, 2017; Paligorova and Santos, 2017).

Noteworthily, there are potential values for examining the two linkages in one unified empirical framework. But, I worry the research contributions are somewhat limited given the presented results. Instead of considering the risk-taking channel (of either bank liquidity or monetary policy), in my view, it would be more interesting (perhaps promising) to focus on the transmission mechanism of the risk-taking channel of monetary policy through bank liquidity. However, the results reported in the current study are preliminary and insufficient (only columns 4-6 in Table 3 and some robustness results in part of Table 4).

In light of the existing studies, the authors should thoroughly explain how their research relates to and contributes beyond the findings documented in the literature. While the authors obviously are aware of these studies (many are cited in the paper), they have made no effort to justify the paper’s distinct contributions. For example, when developing Hypothesis 1, the authors say that “our first hypothesis is consistent with the prediction of Acharya and Naqvi(2012) and Khan et al.(2017)”. If so, what are the values of reexamining this linkage?

Besides the studies listed above, the authors should also check Nguyen and Boateng (2015), which examine the risk-taking behaviour of Chinese banks and its associations with monetary policy and involuntary excess reserves. In addition, instead of being relatively vague when motivating the research questions, I would suggest the authors highlight the importance of understanding the risk-taking channel of monetary policy in association with bank liquidity.

Other comments

• Is model (2) mainly used in Table 2, which tests the association between bank liquidity and risk taking? If so, the authors should remove MP from the model.

• Again, in model (2), Liquidity is given a one-year lag. To make the results more comparable, MP should also be treated with a lag related to the dependent variable banks’ Z-score.

• I don’t understand why the standalone Liquidity is missing in model (3). So the right set of variables should include MP, Liquidity, and MP×Liquidity. In Table 3, the authors remove Liquidity when MP×Liquidity is added (columns 4-6). Why?

• There are some inconsistencies in the tests. For instance, three variables (i.e. reserve, interbank, M2) are used to proxy for the monetary policy. But M2 is not used in the robustness tests in Table 4.

• After showing the robust results of the liquidity risk-taking of monetary policy in Table 4, the study re-test the linkage between risk taking and monetary policy (columns 1-3 of Table 3) using the mediation model. This is rather confusing as I think the paper should extend the analysis related to Hypothesis 2.

 

References

Acharya, Viral, and Hassan Naqvi, 2012, The seeds of a crisis: A theory of bank liquidity and risk taking over the business cycle, Journal of Financial Economics 106, 349–366.

Angeloni, Ignazio, Ester Faia, and Marco Lo Duca, 2015, Monetary policy and risk taking, Journal of Economic Dynamics and Control 52, 285–307.

Chen, Minghua, Ji Wu, Bang Nam Jeon, and Rui Wang, 2017, Monetary policy and bank risk-taking: Evidence from emerging economies, Emerging Markets Review 31, 116–140.

Khan, Muhammad Saifuddin, Harald Scheule, and Eliza Wu, 2017, Funding liquidity and bank risk taking, Journal of Banking and Finance 82, 203–216.

Nguyen, Vu Hong Thai, and Agyenim Boateng, 2015, An analysis of involuntary excess reserves, monetary policy and risk-taking behaviour of Chinese Banks, International Review of Financial Analysis 37, 63–72.

Paligorova, Teodora, and João A.C. Santos, 2017, Monetary policy and bank risk-taking: Evidence from the corporate loan market, Journal of Financial Intermediation 30, 35–49.

Rokhim, Rofikoh, and In Min, 2020, Funding Liquidity and Risk Taking Behavior in Southeast Asian Banks, Emerging Markets Finance and Trade 56, 305–313.

6. PLOS authors have the option to publish the peer review history of their article (what does this mean?). If published, this will include your full peer review and any attached files.

Reviewer #1: No

Reviewer #2: No

---

## [Author Response · Author response to Decision Letter 0]

11 Oct 2022

Response to Journal Requirements:

1.We have ensure our manuscript meets PLOS ONE's style requirements.

2.We have removed the funding information from our manuscript and we don’t need to update our Funding Statement in the Funding Statement section of the online submission form.

3.The role of the funders took in the study is stated as follows: "The funders had no role in study design, data collection and analysis, decision to publish, or preparation of the manuscript."

Response to Reviewer #1:

We appreciated the constructive criticism and suggestion of Reviewer #1. We addressed all the points raised by the reviewer, as summarized below.

1.The concept of liquidity risk-taking channel of monetary policy has not been used in the banking literature. Khan et al.(2017) and Acharya and Naqvi(2012) just investigate the relationship between liquidity and bank risk without monetary policy, while Borio and Zhu (2012) and Chen et al. (2017) focus on the relationship between monetary policy and bank risk, and bank liquidity is treated as a control variable. According to the referee’s suggestion, we strengthen our gap analysis to further explain how our research relates to and contributes beyond the findings documented in the literature. 

2.According to the referee’s advice, we present a graph of the changes in bank liquidity about the Chinese banking sector during the observation period (2003-2018) in the Introduction section.

3.Thanks to the referee’s comment, we strengthen our gap analysis in the Introduction section and Hypothesis development section 

4.To verify the robustness of the results, we use Loans to Deposits ratio as alternative measures of liquidity risk to run robustness checks.

5.We improve the analysis of the estimated coefficients of the interaction term and control other bank-specific characteristics and Crisis dummy indicator in the regressions.

6.According to the referee’s suggestion, we further explain the purpose of using the mediation model and extend the analysis of the estimation results.

7.Thanks to the referee’s comment, we further formulate policy implications for bank managers and regulators.

Response to Reviewer #2:

We want to thank referee #2 for constructive and insightful criticism and advice. We addressed all the points raised by the reviewer as summarized below.

1.According to the referee’s suggestion, we focus on the transmission mechanism of the risk-taking channel of monetary policy through bank liquidity and extend the analysis related to Hypothesis 2.

2.We strengthen our gap analysis in the Introduction section and Hypothesis development section to thoroughly explain how our research relates to and contributes beyond the findings documented in the literature.

3.According to the referee’s suggestion, we remove MP from the model (2) .

4.Considering the possible endogeneity of Liquidity, we adopt its first lag value in the regressions. However, as the MP variables are treated as exogenous, we haven’t treated MP with a lag related to the dependent variable banks’ Z-score.

5.Thanks to the referee’s comment, we reset the model (3) by including MP, Liquidity, and MP×Liquidity.

6.To be consistent in the tests, we use three variables (i.e. reserve, interbank, M2) to proxy for the monetary policy in each models.

7.According to the referee’s suggestion, we further explain the purpose of using the mediation model and extend the analysis of the estimation results related to Hypothesis 2.

---

## [Decision Letter · Decision Letter 1]

8 Dec 2022

Bank liquidity and the risk-taking channel of monetary policy : An empirical study of the banking system in China

PONE-D-22-18762R1

Dear Dr. Zhuang,

We’re pleased to inform you that your manuscript has been judged scientifically suitable for publication and will be formally accepted for publication once it meets all outstanding technical requirements.

Kind regards,

Daphne Nicolitsas

Academic Editor

PLOS ONE

Additional Editor Comments (optional):

Reviewers' comments:

Reviewer's Responses to Questions

**Comments to the Author**

1. If the authors have adequately addressed your comments raised in a previous round of review and you feel that this manuscript is now acceptable for publication, you may indicate that here to bypass the “Comments to the Author” section, enter your conflict of interest statement in the “Confidential to Editor” section, and submit your "Accept" recommendation.

Reviewer #1: All comments have been addressed

2. Is the manuscript technically sound, and do the data support the conclusions?

Reviewer #1: Yes

3. Has the statistical analysis been performed appropriately and rigorously? 

Reviewer #1: Yes

4. Have the authors made all data underlying the findings in their manuscript fully available?

Reviewer #1: Yes

5. Is the manuscript presented in an intelligible fashion and written in standard English?

Reviewer #1: Yes

6. Review Comments to the Author

Reviewer #1: The manuscript titled: Bank liquidity and the risk-taking channel of monetary policy: An empirical study of the

banking system in China, has been revised to address the suggestions of the two reviewers. The authors have been able to appropriately address all the recommendation and improved the quality of the manuscript. However, few minor issues remained unsolved as follows:

1. The contributions of this paper are still not clear enough in both the Abstract and Introduction (the contribution includes of proposing a new concept, developing a new indicator system or developing a new method, etc.)

2. Formula (1) and formula (2) are unreadable. Please check the math expression.

3. Equation (2) on page 13 should include lag (1) and lag (2) of the dependent variable (Risk). However, the equation shows lag (1) twice. Same mistake with equation (3), (4) and (6).

4. The subscripts j represents the province where bank i allocates. However, none of equations includes this subscript.

5. Tables should be placed in the text after the first mentioning/reference to the respective table to make the reader easy to follow the discussions and interpretations of the results illustrated in that table.

6. The conclusion is a bit general, or it lacks distinction. Please refine this further to be more thought-provoking.

7. PLOS authors have the option to publish the peer review history of their article (what does this mean?). If published, this will include your full peer review and any attached files.

Reviewer #1: No

---

## [Editor Report · Acceptance letter]

15 Dec 2022

PONE-D-22-18762R1 

Bank liquidity and the risk-taking channel of monetary policy : An empirical study of the banking system in China 

Dear Dr. Zhuang:

I'm pleased to inform you that your manuscript has been deemed suitable for publication in PLOS ONE. Congratulations! Your manuscript is now with our production department. 

Kind regards, 

on behalf of

Dr. Daphne Nicolitsas 

Academic Editor

PLOS ONE